# Artificial Intelligence-Based Prognostic Model for Urologic Cancers: A SEER-Based Study

**DOI:** 10.3390/cancers14133135

**Published:** 2022-06-26

**Authors:** Okyaz Eminaga, Eugene Shkolyar, Bernhard Breil, Axel Semjonow, Martin Boegemann, Lei Xing, Ilker Tinay, Joseph C. Liao

**Affiliations:** 1Department of Urology, Stanford University School of Medicine, Stanford, CA 94305, USA; shkolyar@stanford.edu (E.S.); jliao@stanford.edu (J.C.L.); 2Faculty of Health Care, Hochschule Niederrhein, University of Applied Sciences, 47805 Krefeld, Germany; bernhard.breil@hs-niederrhein.de; 3Prostate Center, Department of Urology, University Hospital Muenster, 48149 Muenster, Germany; axel.semjonow@ukmuenster.de (A.S.); martin.boegemann@ukmuenster.de (M.B.); 4Department of Radiation Oncology, Stanford University School of Medicine, Stanford, CA 94305, USA; lei@stanford.edu; 5Department of Urology, Marmara University School of Medicine, Istanbul 34854, Turkey; itinay@marmara.edu.tr

**Keywords:** surveillance management, machine learning, artificial intelligence, urologic cancers, data-driven solution, survival modeling

## Abstract

**Simple Summary:**

We describe a risk profile reconstruction model for cancer-specific survival estimation for continuous time points after urologic cancer diagnosis. We used artificial intelligence (AI)-based algorithms, a national cancer registry data, and accessible clinical parameters for the risk-profile reconstruction. We derived a risk stratification model and estimated the minimum follow-up duration and the likelihood for risk stability in prostate, kidney, and testicular cancers. The estimated follow-up duration was in alignment with recognized clinical guidelines for these cancers. Moreover, the estimated follow-up duration was differed by the cancer origin and the disease dissemination status. Overall, the reconstruction of the population’s risk profile for the cancer-specific prognostic score estimation is feasible using AI and has potential application in clinical settings to improve risk stratification and surveillance management.

**Abstract:**

Background: Prognostication is essential to determine the risk profile of patients with urologic cancers. Methods: We utilized the SEER national cancer registry database with approximately 2 million patients diagnosed with urologic cancers (penile, testicular, prostate, bladder, ureter, and kidney). The cohort was randomly divided into the development set (90%) and the out-held test set (10%). Modeling algorithms and clinically relevant parameters were utilized for cancer-specific mortality prognosis. The model fitness for the survival estimation was assessed using the differences between the predicted and observed Kaplan–Meier estimates on the out-held test set. The overall concordance index (c-index) score estimated the discriminative accuracy of the survival model on the test set. A simulation study assessed the estimated minimum follow-up duration and time points with the risk stability. Results: We achieved a well-calibrated prognostic model with an overall c-index score of 0.800 (95% CI: 0.795–0.805) on the representative out-held test set. The simulation study revealed that the suggestions for the follow-up duration covered the minimum duration and differed by the tumor dissemination stages and affected organs. Time points with a high likelihood for risk stability were identifiable. Conclusions: A personalized temporal survival estimation is feasible using artificial intelligence and has potential application in clinical settings, including surveillance management.

## 1. Introduction

Genitourinary malignancies are amongst the most common cancers [1] and their behavior ranges from indolent to lethal, as reflected by a variety of risk profiles [2,3,4]. Treatment decisions and surveillance are informed by risk assessment and stratification [5,6,7,8]. When treatment is applied, patients are followed for a predefined period to identify potentially actionable disease recurrence or progression, as well as to monitor for treatment-associated side effects [9,10]. Nevertheless, stratifying by cancer-specific mortality risks after adequate treatment remains challenging [9,10]. Current risk-profiling strategies are primarily based on results from statistical analyses that include survival probabilities, risk, odds, or hazards ratios [11,12,13,14,15,16]. Clinical practice for the inference of disease risk considers clinicopathological parameters (tumor stage and grade); these parameters are generally reported following consensus guidelines and recommendations (e.g., AJCC staging system [17]); this practice relies primarily on risk stratification at specific time points (i.e., second or fifth year). However, statistical analyses determining this risk stratification are limited by excluding covariates or by following the proportional hazards assumption for survival modeling with covariates [18]. Current methods inadequately account for the time elapsed since cancer diagnosis and, consequently, the dynamic nature of risk profiles [19], which is not nonlinear and varies by time lapse (e.g., the variation of the mortality risks between second and fifth year).

Different ML approaches have been introduced for survival modeling; for instance, Random Survival Forests is a popular non-parametric ML approach adapted to censored survival data [20]. Another approach called DeepSurv is a deep multilayer perceptron that estimates the relative hazards under the familiar assumption of constant baseline hazard as alternative solutions for linear regression or survival forest [21]. DeepHit is a recurrent neural network that involves learning the joint distribution of all event times by jointly modelling all competing risks and discretizing the output space of event times [22]. Recently, Nagpal et al. introduced Deep Survival Machines, a graphical neural network, to estimate the survival distribution and, accordingly, the time-to-event risk [23]. While these approaches are deemed useful for survival modeling, these approaches are, however, not intended to reconstruct the risk profile, nor to determine the surveillance management.

To address the limitations of current methods, we propose a survival model that reconstructs the cancer-specific mortality risk profile of a representative population over an extended period. Our hypothesis is that memorable machine learning (ML) facilitates a time series of cancer-specific survival estimations based on information obtained around the time of cancer diagnosis. This hypothesis is supported by prior studies demonstrating the potential of ML for survival analysis [22,24,25]. We assume that the treatment strategy and disease progression as a result of any oncologic events or conditions have been indirectly priced into the longitudinal risk profile of the representative population. To this end, we applied ML to develop and validate a dynamic risk-stratification model for genitourinary malignancies and conducted simulation studies to evaluate model performance for prostate, kidney, and testicular cancers.

## 2. Materials and Methods

The longitudinal risk profile for cancer-specific mortality was constructed based on clinicopathological information obtained around the time of diagnosis. The term longitudinal refers to the time elapsed since diagnosis. Using the Surveillance, Epidemiology and End-Results (SEER) database (version 18), we identified patients diagnosed with genitourinary cancers between 1975 and 2017 [26]. The SEER registry represents a large representative body of cancer outcomes, allowing for generalizability of findings, making it an excellent source for reconstructing cancer-specific risk profiles of the population [27].

The reconstruction of risk profiles from the representative national cancer registry necessitates assumptions about the population; thus, five assumptions regarding population-based risk profiles for cancer-specific mortality were employed to ease the complexity of survival modeling:(1)Standard-of-care treatment is accessible and utilized;(2)Cancer-specific survival is not impacted by the definition of the standard-of-care treatment, as shown by previous studies [28,29,30,31]. For instance, the selection of available treatment options (e.g., radiation, surgery) for patients sharing similar clinicopathological information marginally alternates cancer-specific survival and, therefore, its effect is negligible at the population level;(3)Cancer-specific survival in the representative populations indirectly incorporate the risk of any significant oncologic event (e.g., failure of the initial treatment, recurrence) or a consequence of an oncologic condition (e.g., developing bone fracture in the setting of metastatic prostate cancer);(4)Cancer-specific survival estimates reflect the cancer-specific risk profile of a population with urologic cancers, given that exact life tables are not readily available [32];(5)Change in cancer-specific mortality risk over time (risk velocity) defines cancer progression risk, while differences in time to progression do not predict cancer-specific survival [33].

### 2.1. Model Development for Mortality Risk-Profile Reconstruction

PROGRESS recommendations for model development and validation were adhered [34]. The SEER database was used to identify patients diagnosed with genitourinary malignancies; this yielded approximately 1.9 million patients, allowing assignment of 90% of patients to a development set and 10% to a test set. Data regarding tumor type, extent, staging, grading, biomarkers if applicable, age at diagnosis, and time from diagnosis in years were obtained. The cancer-specific mortality status at a given time point was also collected. Further information about the input variables can be found in Appendix A.

We applied a simple recurrent neural network (RNN), a memorable machine learning model [35], to reconstruct the risk profile for cancer-specific mortality (risk profile). Here, right-censoring was used for model development [36]. The grid search was applied to find the optimal hyperparameter configuration for the model, as described in the Appendix A. We considered time (*T*) in the feature input of the model in addition to other clinicopathological covariates (*X*). As part of data preprocessing, the development set was sorted by date of diagnosis to account for changes in data patterns and risk profiles with diagnosis time. The final steps of training included the most recent cases to weight towards recent cohorts in every epoch, thus reflecting contemporary treatment strategies and population risk profiles.

For all patients, the true endpoint was cancer-related death status, and the model output was the prediction score for cancer-specific mortality. We expected the prediction scores not to reflect the survival probability due to overconfidence bias [37] and the inverse correlation between prediction score and survival probability. Accordingly, we re-calibrated the prediction scores to closely correlate with the Kaplan–Meier (KM) survival estimates (survival probability) on the development set and fixed the calibration factor during the test. Estimating the time series of true survival probabilities in the population as a reference for model calibration was feasible given the large cohort size (more than one million patients) of the development set [38]. Details of hyperparameter configuration for model development and calibration for cancer-specific mortality risk estimation can be found in Appendix A.

### 2.2. Evaluation Metrics

We measured the discriminatory accuracy by Harrell’s Concordance Index [39]. The utility of survival modeling was evaluated by the capability to define discriminative risk groups. Here, we assessed the observed cancer-specific survival probabilities at the 10th year from diagnosis after stratifying the test set by quantile thresholds (10%, 50%, 90%), determined from the training set for the prognosticated cancer-specific death probability. The log-rank test determined the discrimination significance between the two risk groups. The two-sided statistical significance is reached when *p* < 0.05.

### 2.3. Exploratory Studies for Potential Utility

#### 2.3.1. Algorithms to Measure the Risk Velocity

We considered measuring changes in the time series for survival probabilities to quantify the dynamic nature of the cancer risk profile. After the model development with the best hyperparameter configuration and its calibration, we estimated risk velocity as the differences in survival probabilities between two time points. We defined time points and calculated the risk velocity as follows:
The collection of the survival probabilities (*S*) is defined as S={s |s∈ ℝ ∧ 0≤s≤1}.We defined a set of 120 time points (*P)* for survival probabilities; we estimated the years elapsed (time lapse, *f*) from the age at diagnosis (*a*) to the median U.S. life expectancy [40] (*b*) (78 years); the time distance (*k*) between two time points was determined after dividing the time lapse by 120 time points.
F={ f | f∈ ℕ+ ∧f(a,b)={b−a, a<b 1 , a≥b , a,b ∈ ℕ+}k=f120 where f ∈F and k ∈ ℝ+P={p∈S | |P|=120 } and A={kn}n=0119 where *k* is a constant from ii and *A* is a set of time positions for survival probabilities *P*, thereby P⟺A.

To illustrate this, we provide the following example: A patient is diagnosed with a urologic cancer at 66 years of age. We obtain a period (*f*) of 12 years and the time distance (*k*) between two time points corresponds to 0.1 year (1.2 months). We then assign to each time point (*p*) a survival probability (*s*) determined by our model. After repeating this calculation for each time point, a total of 120 time points (*P*) with the corresponding survival probabilities (*S*) are generated.

2.The timeline created by time points (*P*) was then divided into intervals (*I*), thereby resulting in sequences of interval-associated time points with survival probabilities. The interval length (*l*) is dynamic and corresponds to the half number of time points divided by the time lapse (e.g., for a time lapse of 12 years, the interval length would be 5 time points or 6 months). The total number of intervals (*n*) is the total time points (120) divided by the interval length. The following algorithm to define the intervals is adjusted according to the time lapse to facilitate the interval scaling:
ivl=[1202f]=[60f] where f ∈F ∧ l ∈ ℕ+vn=[120l] where n ∈ ℕ+viI={[li,l(i+1)]}i=0n−1 where l is a constant and calculated using iv.
3.After generating the intervals (*I*), we estimated the risk velocity (v) per interval. The risk velocity (*v*) is the difference between the first (*s_p_*_1_) and the last (*s_pl_*) survival probabilities [*s_p_*_1_, *s_pl_*] of a time interval from *I*.
vii$v=|s_{p1}-s_{pl}|$

For instance, the risk velocity for the first interval with 5 time points (6 months) would be the absolute difference between the survival probability of the first point (*s_p_*_1_) and the survival probability of the fifth time point (*s_p_*_5_). After the velocity calculation for all intervals, a quantitative assessment of the dynamic of the risk profile can be performed.

#### 2.3.2. Simulation Study

We conducted a simulation study with the assumption that patients were receiving standard of care therapy. The simulation covered 10,606,996 random and unique clinical conditions based on the TNM stage, biomarker status, gender, and age at diagnosis for three urologic cancers (prostate, kidney, and testis). For all cases, the time lapse from diagnosis was set to 12 years. We explored two utilization scenarios of the novel model, which are:
(1)Recommender for the minimum follow-up duration

We developed the risk velocity-based follow-up duration assessment as the following; the definition of the period for close follow-up is based on the grade of velocities; time intervals with score velocities exceeding the threshold of 0.5% were marked for a close follow-up. This threshold was used because the overall lowest 5-year cancer-specific death probability was 0.5% [41]. For each cancer, bar and area plots were applied to visualize the distributions of the recommended follow-up durations for all simulation cases or stratified by the tumor dissemination status; data are presented at 25% (low/first quartile), 50% (median) and 75% (upper/third quartile) percentiles of the recommended results. The lower quartile corresponds with the minimum recommended follow-up duration.
(2)Determine intervals with unchanged risk profiles


Building on the previous scenario, we determined risk stabilization based on the risk velocity. A risk velocity of 0 indicates that the risk profile remained unchanged or stable for a given time interval, while a risk velocity deviating from 0 implies that the risk condition is changing. A high velocity suggests a rapid change in the risk profile over time, and a low velocity suggests that the risk profile changes slowly. After calculating the velocities during 12 years after diagnosis, as described previously, we estimated the frequency of nil velocity in each year for all simulation cases or after stratification by the tumor dissemination status. The results were plotted as a bar graph over 12 years. Years where a nil risk velocity was found in more than 30% of the year cases were then highlighted. The threshed of 30% was aimed to visually distinguish the years according to the frequency of nil velocity. Given the large simulation cohort, the *p*-value estimation was omitted for frequency comparison.

## 3. Results

Using the SEER database, complete follow-up information was available for 1,941,893 cases. Here, 14.34% of patients died due to one of the urologic cancers, with an overall median follow-up of 12 years (range 1–44 years). When patients were stratified by age at diagnosis, the age group of 30–64-year-olds had a median follow up of 14 years, and the age group of 65–69-year-olds had a median follow up of 13 years (Appendix A). By stratifying according to the cancer-specific death status, the median time to cancer-specific death was 11 years, and the median observation time was 13 years for those who were event-free at the time of the 18th version of the SEER database.

### 3.1. Model Development and Evaluation

The grid search analysis for hyperparameter configuration revealed that it is optimal to select simple recurrent units (sRNN) and RMSprop for model designs. The number of units of RNN was set to 16. The grid search analysis did not recommend the inclusion of the dropout function. The model was trained on a training set (*n* = 1,747,703 cases, from which 250,555 cases died because of urologic cancers); during model training, the last 10% of the training set (*n* = 17,477) was excluded from the model training and used for in-training validation. The test set had 194,190 cancer patients, from which 27,840 patients died due to urologic cancers. The overall concordance index on the test set was 0.80 (95% CI: 0.795–0.805). The Kaplan–Meier (KM) curve revealed a well-fitted model for cancer-specific survival rates over a follow-up period of 35 years on the test set (Figure 1). The overall delta difference was 0.008 (0.08%), which indicates a perfect match between the predicted and real survival estimates over 35 years on the test set. When the test set was stratified by the cancer type, we found that the KM fitness mostly remained unchanged (range: 0.004–0.01).

The stratification by quantile thresholds showed well-discriminative risk groups (Log-rank *p* < 0.001). When compared to the tumor dissemination status used by SEER for survival estimation (Figure 2), we identified that our model could introduce a very-low risk group with a 10-year cancer-specific survival (CSS) rate of 99.55% (95% Confidence interval-CI-: 99.43–99.64). In contrast, the best 10-year CSS rate by the tumor dissemination status was for localized cancer diseases (CSS rate: 95.92%; 95% CI: 95.72–96.11). The 10-year CSS rate in distant cancer was 61.72% (95% CI: 60.75–62.67), whereas the 10-year CSS rate was 66.72% (95% CI: 65.49–66.88) for the high-risk group determined by the predicted probability. This observation indicates that our approach is not impacted by overconfidence bias and the resulting probability estimation reflects the risk groups found in the SEER database.

### 3.2. Exploratory Studies

(1)Recommender for the minimum follow-up duration

Our solution suggested at least a four year follow-up after diagnosis for prostate cancer or five years for kidney and testicular cancers. In the upper quartile (75%) of the whole simulation cohorts, the follow-up durations can reach 11 years for prostate cancers, 8 years for kidney cancers or 7 years for testicular cancers.

When we evaluated the minimum duration of the follow-up suggested by our algorithms after stratifying by the tumor dissemination status, we found that the suggested minimum follow-up periods also differed between localized cancers (4 years) and distant metastases (9 years) for kidney cancers (Figure 3). In contrast, the suggested minimum follow-up periods for prostate and testicular cancers remained unchanged after stratifying by the tumor dissemination status. Other possible follow-up durations can reach up to 10 years for prostate cancers in 75% of simulated cases, regardless of the tumor dissemination status. In kidney cancers, 75% of simulated cases can have a maximum follow-up duration of 9 years, regardless of the tumor dissemination status. Finally, the simulation study revealed that follow-up can continue up to 8 years for localized testicular cancers or 9 years for testicular cancers with distance metastasis.

(2)Determine intervals with unchanged risk profiles

As shown in Figure 3, the likelihood for risk stabilization (no change in the cancer-specific mortality risk) varied by the years after diagnosis for the major three urologic cancers. The cancer-specific mortality risk was more likely (>30%) stabilized in the first two years after diagnosis for prostate cancer compared to the remaining years; the likelihood for stabilized cancer-specific mortality risks was higher in the third and fourth years for kidney cancer than in the remaining years. For testicular cancers, the period between the fourth year and the seventh year revealed the highest likelihood for stabilized cancer-specific death risks.

The distribution of stabilized cancer-specific mortality risk in the years after diagnosis was mostly comparable between localized cancers and distant metastases in prostate cancer. The cancer-specific mortality risk was more likely stabilized in the third year for local cancers or in the third and fourth years for distant metastases of kidney cancers when compared to other follow-up years. Finally, the cancer-specific mortality risk for localized testicular cancers was more likely stabilized during the 3rd, 4th, and 6th years in comparison to the remaining years, whereas the highest likelihood for stabilized risk profile was found during the period from 5th to 7th year in testicular cancers with distant metastases.

## 4. Discussion

This study introduces and validates the risk-profile reconstruction concept of a population using advanced machine learning to estimate the cancer-specific survival probability. Our prognostic model can accurately estimate the cancer-specific mortality risk and provide a discriminative risk stratification. Furthermore, we delivered a well-calibrated model for cancer-specific survival estimation over a long period, while providing stability and good fitness [42]. Our cancer-specific prognostic model was further developed by feeding data sorted by date of diagnosis (from 1975 to 2017) to counter the concept drift, a known problem in machine learning [43,44], that can be caused by changes in the data patterns and risk profiles over the years due to advancement in clinical strategies. In addition to handling concept drift, the assumptions on the national cancer registry define hidden features of the cancer registry database for survival modeling.

Prostate, kidney, and testicular cancers are amongst the most common cancers seen in urologic oncology. Additionally, kidney cancers affect both genders and testicular cancers are amongst the most frequent cancer diseases in young males [45], facilitating the model assessment on heterogenous simulation cohorts. We identified that the risk velocity for cancer-specific mortality is a useful parameter for developing algorithms aimed at finding optimal observation intervals after cancer diagnosis. Moreover, the minimum observation intervals for three major urologic cancers suggested by our algorithms covered the first 5 years mostly considered by the current clinical guidelines and the follow-up plan practices [46,47,48]; our finding indicates the current follow-up plan designed by expert opinions aims to meet the minimum follow-up requirement for the surveillance management; thus, certain cases with high cancer-specific mortality risks mandate additional adjustment of the current surveillance strategy according to the risk profile, as our simulation study reveals. The risk profile appeared to be stabilized in the first three years and at the fifth year. This observation aligns with the current clinical practice that recommends a modification of the surveillance schedule during the third year and/or the fifth year for these major urologic cancers [46,47,48]. In current clinical practice, the first two or three years of follow-up generally include frequent follow-up visits and aim at capturing treatment complications or cancer residuals earlier; follow-ups beyond the second or third year after diagnosis are generally intended to monitor the oncologic condition of survivors who have received or are receiving standard-of-care treatment. Our simulation further reveals that testicular cancers with distant metastases reach risk stability later than prostate and kidney cancers, or localized testicular cancers; this late occurrence of the risk stability is presumably driven by the biology of metastases in testicular cancers [49,50] and the delayed reflection of treatment effects into the risk velocity. Nevertheless, further in-depth study regarding risk stability is required to validate our assumptions. Overall, these observations support that the conclusion of our solution derived from the risk velocity is clinically plausible and aligned with expert opinions.

Recently, studies on survival modeling using deep learning have emerged [22,51,52,53,54], and mostly continue to respect the proportional hazards assumption [18] or apply a predictive model at fixed time points. A notable similar study to ours conducted by Lee et al. introduced a novel model called DeepHit aimed to overcome the limitation of current statistical analyses by learning the distribution of survival times directly [22]. Although DeepHit is a novel solution, this study was limited by a lack of calibration and by only considering discriminative accuracy (c-index) to support their conclusion; c-index is not enough to determine how well the model learned the distribution of survival times, nor is it a suitable measurement to assess the model performance at different time points [55,56]. In contrast, our study demonstrated discriminative accuracy and risk profiling of the population over time to illustrate the time-dependent nature of the model assessment. Our findings further indicate that the resulting cancer-specific probabilities are well-calibrated and not impacted by overconfidence, a major issue in deep learning [37]. Due to the well-calibrated model, we can identify distinguishable risk groups using the predicted cancer-specific probabilities. For some cancer types, the model fitness has declined 26 years after cancer diagnosis, possibly due to the decrease in the sample size after 26 years. Finally, we examined the potential clinical utility (clinical significance) of our approach using simulation tests that incorporated different conditions of urologic cancer diseases and delivered results in accordance with current clinical practice for follow-up management.

The current study has limitations that warrant mention. First, this is a proof of concept for the development of a sophisticated model with potential utility in surveillance management. Future studies will focus on evaluating the cost-effectiveness of surveillance planning and further adoption of this concept to clinical needs before conducting a prospective study. Although the SEER database does not include detailed treatment information, the version of the SEER database (1975–2017) considered by the current study very likely covered patient groups who received standard-of-care treatment that evolved over time. Diagnostic and therapeutic improvements have helped to detect cancers at earlier stages, resulting in longer survival, as reflected in the SEER database [41]. Most recent drug options for various urologic malignancies found their first approval in early 2000 [57,58]. The staging modalities were not standardized, which may have resulted in an under- or over-estimation of cancer extent and thereby influenced the data quality. The SEER data, however, is likely a good representative of general clinical practice. Furthermore, the SEER database is the only comprehensive population-based database in the United States and represents an excellent resource for studying the survival of patients diagnosed with cancer [41]. The SEER program considers standards for data quality in national cancer registries [59,60]; the SEER database covers more heterogenous cases than any data collected from a single institution and is therefore ideal for a generalizable model validation [27]; the database is useful in defining the utilization boundary of models as this database reflects real-world challenges (e.g., different patient generations). We considered the parameters that are prognostic, clinically widely adapted and sufficient for model development for accurate risk-profile reconstruction. We considered tumor staging and dissemination status instead of tumor grading for kidney or bladder cancers due to the significant variation in the grading definitions and their significant impact on cancer-specific mortality [61,62]. We considered serum AFP levels in our survival modeling because AFP is a surrogate biomarker for non-seminoma testicular cancers [63]. Although the histological subtypes were not considered, our current study demonstrated the usability of the limited information to develop a sophisticated survival model. We did not train numerous models in parallel due to computational limitations and the results indicate our method is sufficient for model selection.

In summary, the current work introduces, as proof-of-concept, a dynamic risk profile model using advanced machine learning and the risk-profile reconstruction from a population-based database; the novel survival model provides a foundation for a time series of risk estimations and velocity calculation that have potential application in shaping the follow-up plans for cancer survivors.

## 5. Conclusions

This study reveals a feasible data-driven artificial intelligence solution for cancer-specific survival estimation and potentially the follow-up management of urologic cancers.

## Figures and Tables

**Figure 1 cancers-14-03135-f001:**
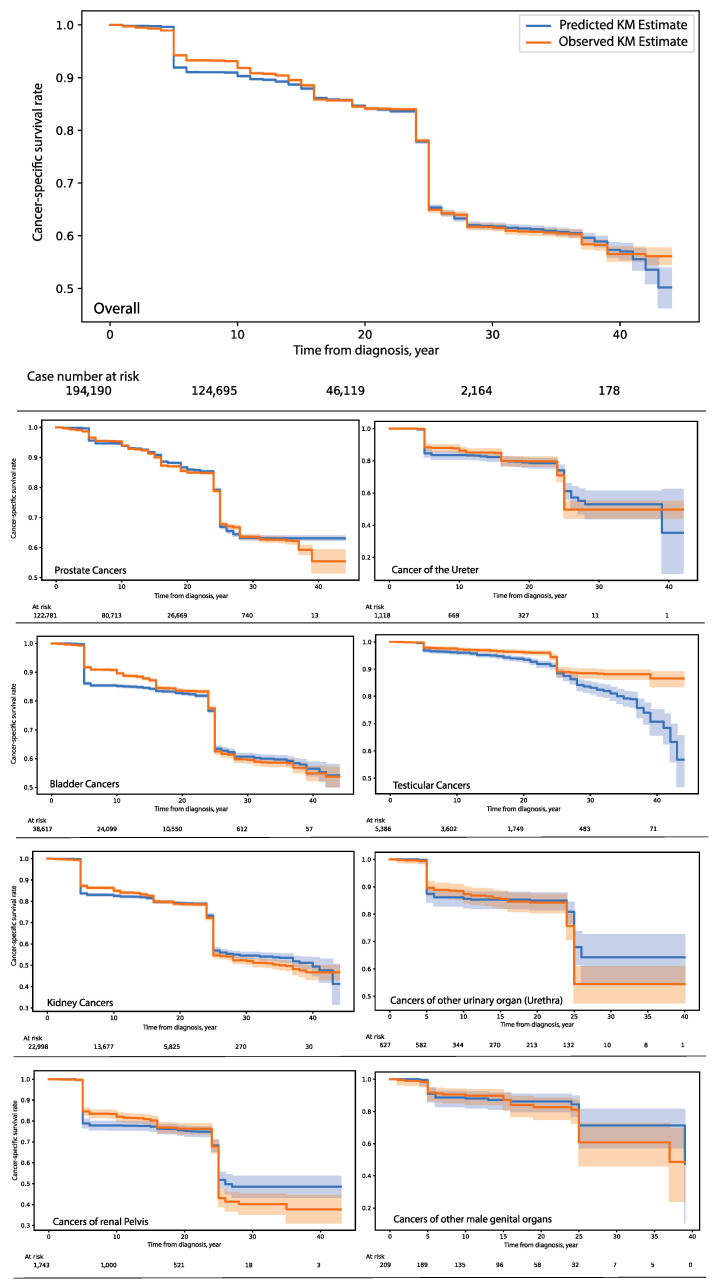
Shows how well the predicted Kaplan–Meier (KM) estimates fit to the observed KM estimates even after stratifying by cancers of different organs on the disjointed test set after the model calibration on the training set. A good overlap means a well-fitted model that facilitates the reconstruction of the unseen population’s survival history with cancer diseases. 95% Confidence intervals are visualized as bands around KM curves. For most cancers, we observed a max deviation of 5.5% between predicted and observed survival estimates for times lapse of 26 years after diagnosis. The model predictions become less reliable after 28 years from diagnosis for testicular cancers.

**Figure 2 cancers-14-03135-f002:**
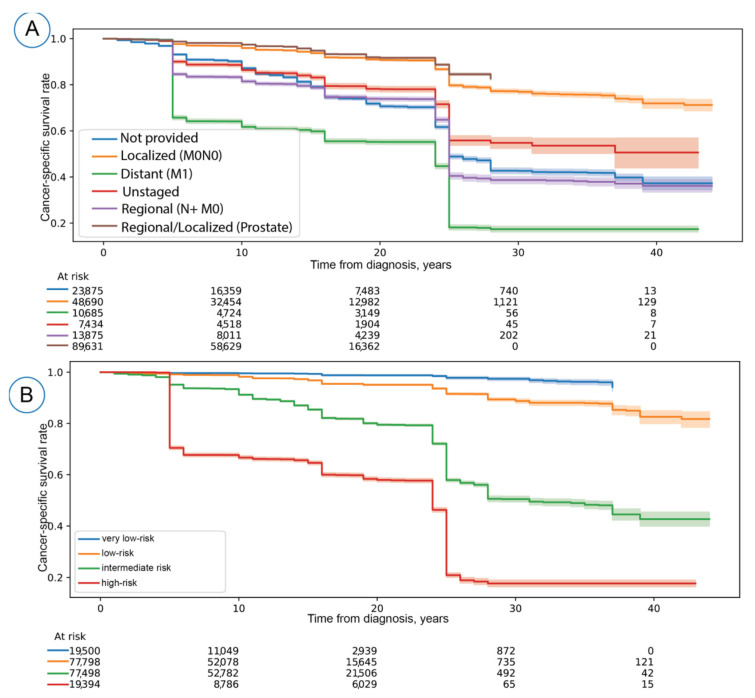
(**A**) the Kaplan–Meier (KM) Curves for the SEER staging groups and (**B**) KM curve after stratifying the prognosticated survival probabilities using quantile thresholds (10%, 50%, 90%) defined on the training set. The KM curves form A and B were significantly discriminative (Log-rank *p* < 0.005). For (**A**), we calculated Log-rank P using localized, distant, regional, and regional/localized stage groups. Localized and regional prostate cancer cases were combined due to their high 5-year survival rates (nearly 100%) compared to other entities. The KM curves were generated on the test set. 95% Confidence intervals are visualized as bands around KM curves.

**Figure 3 cancers-14-03135-f003:**
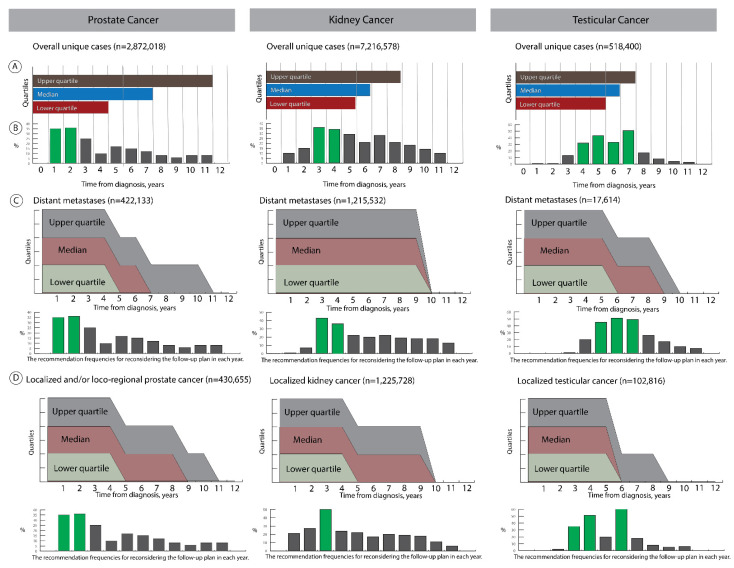
Describes the results of the simulation for the follow-up duration assessment and the risk-profile stability of unique clinical conditions in prostate, kidney, and testicular cancers for all simulation cases (**A**,**B**) and after stratifying according to the tumor dissemination status (**C**,**D**). The lower quartile corresponds with the minimum recommended follow-up duration. The green bars highlight the years more likely have stable cancer-specific mortality risk compared to other years.

## Data Availability

We utilized the existing python packages for the model development and our evaluation. The packages are NumPy 1.18, lifelines 0.24.4, scikit-learn 0.23, TensorFlow 2.2, Keras 2.4. The code for model development and calibration can be obtained from: https://github.com/oeminaga/AIFollowUpUrologicCancer.git.

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
