# Peer review of "Artificial Intelligence-Based Prognostic Model for Urologic Cancers: A SEER-Based Study"

_cancers, 2022, doi:10.3390/cancers14133135_

Round 1

Reviewer 1 Report

Eminaga et al. present an AI-based risk profile reconstruction method in the current manuscript. The method is very interesting and it most certainly has a translational importance as the model helps to decide on the optimal follow-up time for urologic cancers. 

The paper is methodologically solid, however, I have some concerns, mostly related to the data used. 

1. I haven't used the SEER database before and it seems like an impressive resource, although the data presented in the paper seems a bit odd. First of all, the median follow-up times, presented in the supplementary material: is it really the case that the median follow-up is ~12 years for those who were 85+ at the time of diagnosis? This seems rather unrealistic. 

2. The number of cancer-specific (and overall) deaths shows a very odd periodicity, with increased numbers roughly every 5 years. the mortality is 3x higher in year 5 than in the previous years, while in e.g. year 7 virtually nobody dies of cancer. There is a very big jump in mortality around the 25th year, so the cancer-specific mortality in year 25 seems to be higher than in the first 5 years, which I find again very unrealistic. 

Checking the online browser of SEER, this should not be the case, however, they only show the data after 2000. Was there may be a change in methodology? Was the data somehow pooled? 

I would recommend the authors to check the underlying data again and rather lose many, but use the newer records only, or try to test the performance of the model on the newer data. 

Again, I find the accuracy of the prediction impressive, but if the model is based on "wrong" data, the model is not generalizable. To ensure it is not the case, it would be interesting to test the performance on an independent dataset. 

Despite the drop in survival at year 5, the KM-curves based on the simulated datasets don't seem to show this the survival curves look more realistic here. How would the authors explain this? 

Author Response

We would like to thank the reviewer 1 for the valuable comments and feedback. Please see our responses to comments below.

Eminaga et al. present an AI-based risk profile reconstruction method in the current manuscript. The method is very interesting, and it most certainly has a translational importance as the model helps to decide on the optimal follow-up time for urologic cancers. 

The paper is methodologically solid; however, I have some concerns, mostly related to the data used. 

  1. I haven't used the SEER database before and it seems like an impressive resource, although the data presented in the paper seems a bit odd. First of all, the median follow-up times, presented in the supplementary material: is it really the case that the median follow-up is ~12 years for those who were 85+ at the time of diagnosis? This seems rather unrealistic. 

  1. We thank the reviewer for highlighting this concern. The age stratification is defined by SEER without any modification in our work. The SEER program considers any elderly patients equal to or above 85 yrs. as 85+. According to a recent study, the cancer death rate for those patients is low compared to other age groups, and 40% of elderly cancers in 85+ patients are prostate cancer that has a 5-yr. survival rate of 100% [1]. We observed similar follow-up duration between patient groups “80-84” and “+85”, suggesting these patients are similar. We speculate that only the most fit patients in these age groups undergo any diagnostic testing, and that this may account for some of this finding - this discussion is beyond the scope of our study, however.

To mitigate any potential confusion, we modified the caption of Figure S1A to include the definition for 85+.

“85+: equal to or above 85 years. The figure shows a decline in the median follow-up duration with aging.”

  1. The number of cancer-specific (and overall) deaths shows a very odd periodicity, with increased numbers roughly every 5 years. the mortality is 3x higher in year 5 than in the previous years, while in e.g. year 7 virtually nobody dies of cancer. There is a very big jump in mortality around the 25th year, so the cancer-specific mortality in year 25 seems to be higher than in the first 5 years, which I find again very unrealistic. 

  1. We thank the reviewer for highlighting this point. We refer the reviewer to Figure S1B as it provides a normalized per-year cancer death frequency instead of Figure S1C. It shows a declining trend over the years after diagnosis. We reported that the model reliability declines after 26 and 28 years as the model fitness degrades after these time points. This is presumably due to the sample size and the data acquisition strategy for cases with follow-up longer than 25 years.  Nevertheless, we did not observe any strong impact of this sample size issue on KM after stratifying by organs or SEER staging groups.

To mitigate any potential misunderstanding, we excluded Figure S1C from the supplementary section and added the following sentence in Figure S1B.

“Overall, the cancer-related death distribution has a declining trend with time.”

  1. Checking the online browser of SEER, this should not be the case, however, they only show the data after 2000. Was there may be a change in methodology? Was the data somehow pooled? 

  1. A) The SEER program follows a strict quality control that was initiated in 1973. The web portal aims at providing the latest cancer trend and therefore considers the survival estimation from 2000. However, the SEER program did not raise any concerns regarding cases prior to 2000 and these are available for research.

For more clarification, we added the following sentence in “discussion” section, page 11, paragraph 2:

“The SEER program considers standards for data quality in national cancer registries [1,2]”

  1. I would recommend the authors to check the underlying data again and rather lose many, but use the newer records only, or try to test the performance of the model on the newer data.  Again, I find the accuracy of the prediction impressive, but if the model is based on "wrong" data, the model is not generalizable. To ensure it is not the case, it would be interesting to test the performance on an independent dataset. 

  1. A) We thank the reviewer for highlighting this point and acknowledge the limitations of working with a large dataset akin to SEER. Although there are inherent issues with such data acquisition and storage programs, the SEER program is known for its quality control efforts. In order to account for improvements in both treatment and in data acquisition and storage, we weighted more heavily the most recent cases for the risk profile reconstruction as described in the methods section. Despite its limitations, the SEER database is the most comprehensive form of multi-institutional data available and is used frequently for similar applications.

We added the following sentences in “discussion” section, page 11, paragraph 2:

“The SEER database covers more heterogenous cases than any data collected from a single institution and is therefore ideal for a generalizable model validation [3]; the database is useful in defining  the utilization boundary of models as this database reflects real-world challenges (e.g., different patient generations).”

  1. Despite the drop in survival at year 5, the KM-curves based on the simulated datasets don't seem to show this the survival curves look more realistic here. How would the authors explain this? 

  1. A) We thank the reviewers for this comment. KM estimates were performed on the test sets using real cases from SEER database. The significant changes at 5-years are mostly contributed to by highly aggressive cancers with distant metastases or high-risk cancers, as shown in Figure 2.

Reference:

  1. Margaret Adamo, R.; Lewis, D.R.; Peace, S. Revising the Multiple Primary and Histology Coding Rules. Journal of Registry 2007, 34, 81.
  2. program, S. SEER Quality Improvement. Available online: https://seer.cancer.gov/qi/ (accessed on 06/21/2022).
  3. Eminaga, O.; Al-Hamad, O.; Boegemann, M.; Breil, B.; Semjonow, A. Combination possibility and deep learning model as clinical decision-aided approach for prostate cancer. Health informatics journal 2020, 26, 945-962.

Reviewer 2 Report

- The main question is to study the role of machine learning as a time series for cancer-specific survival estimations based on information obtained around the time of cancer diagnosis. They assumed that the treatment strategy and disease progression as results of any oncologic events or conditions have been indirectly priced into the longitudinal risk profile of the representative population.

 the conclusions consistent with the evidence and arguments presented and they address the main question posed, the references appropriate

Author Response

We would like to thank reviewer 2 for recognizing the potential of this study.

Reviewer 3 Report

Eminaga and coworkers have created an AI-based prognostic model for cancer using the SEER database. They present a nice paper, well explained, methodologically strong, well conducted and with interesting results. However, the reviewer has some questions and comments that it would be good if they could address.

1 - It would be good to have more references to similar studies with or without ML in the introduction. Also to the use of ML models in general for forecasting modelling, for example.

2 - To what extent do the authors believe that the initial assumptions may influence the outcome of the study? It would be good to have some lines on this in the conclusions or elsewhere

3 - Did the authors test different types of NNs or other ML methods? It seems that they do from the information in the SI but it would be good to include some more reference to this in the main text. Also to hyperparameter optimisation, etc.

4 - It would be good if the authors showed the model quality metrics in a more visual way with a graph or a table. Also the metrics of the tested, but not selected models.

5 - Why do the authors believe that the model predictions are less reliable after a few years? It would be nice to have a few more lines on this in the text as they do for stabilisation.

6 - The axes in figures S1A and B are difficult to read even when zoomed in. It would be good if they were modified.

Author Response

We would like to thank the reviewer 3 for the valuable comments and feedback.

Please see responses to comments below.

Eminaga and coworkers have created an AI-based prognostic model for cancer using the SEER database. They present a nice paper, well explained, methodologically strong, well conducted and with interesting results. However, the reviewer has some questions and comments that it would be good if they could address.

1 - It would be good to have more references to similar studies with or without ML in the introduction. Also to the use of ML models in general for forecasting modelling, for example.

  1. We thank the review for this valuable comment.

We added the following paragraph in “Introduction” section, page 2, paragraph 2:

“Different ML approaches have been introduced for survival modeling; for instance, Random Survival Forests is a popular non-parametric ML approach adapted to censored survival data [1]. Another approach called DeepSurv is a deep multilayer perceptron that estimates the relative hazard under the familiar assumption of constant baseline hazard as alternative solutions for linear regression or survival forest [2]. DeepHit is recurrent neural network that involves learning the joint distribution of all event times by jointly modelling all competing risks and discretizing the output space of event times [3]. Recently, Nagpal et al introduced Deep Survival Machines a graphical neural network to estimate the survival distribution and accordingly the time-to-event risk [4]. While these approaches deem useful for survival modeling, these approaches are, however, not intended to reconstruct the risk profile nor to determine the surveillance management.”

We added the following sentence (highlighted in yellow) in “Introduction” section, page 2, paragraph 2:

“Our hypothesis is that memorable machine learning (ML) facilitates a time series of cancer-specific survival estimations based on information obtained around the time of cancer diagnosis. This hypothesis is supported by prior studies demonstrated the potential of ML for survival analysis [3,5,6].”

2 - To what extent do the authors believe that the initial assumptions may influence the outcome of the study? It would be good to have some lines on this in the conclusions or elsewhere

  1. The assumptions we considered reflect the properties of national cancer registries. These properties are hidden features of the database and advanced machine learning can capture such hidden features.

We added the following sentence in discussion section, page 9, paragraph 1:

“In addition to handling concept drift, the assumptions on the national cancer registry define hidden features of the cancer registry database for survival modeling.”

3 - Did the authors test different types of NNs or other ML methods? It seems that they do from the information in the SI but it would be good to include some more reference to this in the main text. Also to hyperparameter optimization, etc. It would be good if the authors showed the model quality metrics in a more visual way with a graph or a table. Also the metrics of the tested, but not selected models.

  1. the optimal hyperparameters configuration and model selection was conducted in parallel using Grid search. For this purpose, we considered 10,000 patients from the development set. The best model from Grid search is then trained on the whole training set. So, the negative log loss values are only relevant for model selection, but not to demonstrate the model accuracy as we would need to train all models on the whole training set and then the test set, which is computationally expensive. We believe our selection strategy justifies the model selection and the hyperparameter configuration. We also provide example codes that we used to determine and train the survival   

We added the following sentence in “discussion” section, page 11, paragraph 2:

“We did not train numerous models in parallel due to computational limitation and the results indicate our method is sufficient for model selection.”

5 - Why do the authors believe that the model predictions are less reliable after a few years? It would be nice to have a few more lines on this in the text as they do for stabilisation.

  1. We thank the reviewer for the valuable comment. Our model fitness degrades after 25 years presumably due the declining sample sizes over the years. Accordingly, we added the following sentence in discussion section, page 10 and paragraph 3.

“For some cancer types, the model fitness has declined 26 years after cancer diagnosis, possibly due to decrease in the sample size after 26th years”

6 - The axes in figures S1A and B are difficult to read even when zoomed in. It would be good if they were modified.

  1. This has been updated to allow for easier viewing.

Reference:

  1. Ishwaran, H.; Kogalur, U.B.; Blackstone, E.H.; Lauer, M.S. Random survival forests. The annals of applied statistics 2008, 2, 841-860.
  2. Katzman, J.L.; Shaham, U.; Cloninger, A.; Bates, J.; Jiang, T.; Kluger, Y. DeepSurv: personalized treatment recommender system using a Cox proportional hazards deep neural network. BMC medical research methodology 2018, 18, 1-12.
  3. Lee, C.; Zame, W.R.; Yoon, J.; van der Schaar, M. Deephit: A deep learning approach to survival analysis with competing risks. In Proceedings of the Thirty-second AAAI conference on artificial intelligence, 2018.
  4. Nagpal, C.; Li, X.; Dubrawski, A. Deep Survival Machines: Fully Parametric Survival Regression and Representation Learning for Censored Data With Competing Risks. IEEE J Biomed Health Inform 2021, 25, 3163-3175, doi:10.1109/JBHI.2021.3052441.
  5. Giunchiglia, E.; Nemchenko, A.; Schaar, M.v.d. RNN-SURV: A deep recurrent model for survival analysis. In Proceedings of the International conference on artificial neural networks, 2018; pp. 23-32.
  6. Hao, L.; Kim, J.; Kwon, S.; Ha, I.D. Deep learning-based survival analysis for high-dimensional survival data. Mathematics 2021, 9, 1244.